# Mechanisms Underlying Bone Loss Associated with Gut Inflammation

**DOI:** 10.3390/ijms20246323

**Published:** 2019-12-15

**Authors:** Ke Ke, Manoj Arra, Yousef Abu-Amer

**Affiliations:** 1Department of Orthopaedic Surgery and Cell Biology and Physiology, Washington University School of Medicine, St. Louis, MI 63110, USA; ke.ke@wustl.edu (K.K.); arram@wustl.edu (M.A.); 2Shriners Hospital for Children, St. Louis, MI 63110, USA

**Keywords:** gut inflammation, mucosal immunity, bone loss, risk factors, animal models, osteoclasts

## Abstract

Patients with gastrointestinal diseases frequently suffer from skeletal abnormality, characterized by reduced bone mineral density, increased fracture risk, and/or joint inflammation. This pathological process is characterized by altered immune cell activity and elevated inflammatory cytokines in the bone marrow microenvironment due to disrupted gut immune response. Gastrointestinal disease is recognized as an immune malfunction driven by multiple factors, including cytokines and signaling molecules. However, the mechanism by which intestinal inflammation magnified by gut-residing actors stimulates bone loss remains to be elucidated. In this article, we discuss the main risk factors potentially contributing to intestinal disease-associated bone loss, and summarize current animal models, illustrating gut-bone axis to bridge the gap between intestinal inflammation and skeletal disease.

## 1. Introduction

The intestinal immune system is primarily involved in controlling and limiting mucosal immune responses to enteric and ingested bacterial antigens, through the actions of gut-associated lymphoid tissue (GALT) [1]. GALT is also involved in mounting appropriate immune responses to pathogenic bacteria in order to protect the body. However, inappropriate GALT activity results in a variety of unnecessary or disrupted immune activity which could manifest as functional disorders within or beyond the gastrointestinal (GI) tract [2,3,4]. Inflammatory bowel disease (IBD) is one such systemic, bowel-centric condition with chronic immune activation and inflammation within the GI tract that to this date remains poorly understood. Ulcerative colitis (UC) and Crohn’s disease (CD) are two distinct forms of IBD, both characterized by a multifactorial, incompletely known etiology involving genetic components, environmental factors, and altered gut microbiota (GM) [5]. The extra-intestinal manifestation in IBD patients involves development of disease in other organs, including the joints, skin, or eyes [6], that can lead to significant co-morbidities and loss of quality of life.

Patients with GI disease, especially IBD, are at a significantly higher risk of developing skeletal disease, such as osteoporosis (OP) (17–41% of patients) or osteopenia (22–77% of patients), and other enteropathic arthropathies [7,8,9,10]. OP is a systemic skeletal disease characterized by significantly reduced bone mass/mineral density (BMD) and deterioration of bone microarchitecture [11]. Cases of OP are classified as primary or secondary. Primary OP is the most common and prevalent form, including juvenile, postmenopausal and senile OP. The causes of secondary OP are vast and attributed to underlying diseases, including hematological diseases, GI disease, vitamin deficiency, solid organ transplantation, and chronic medication [12]. The main therapies for treating OP target bone remodeling involve either inhibiting the bone resorbing osteoclasts (OCs) or promoting the activity of bone forming osteoblasts (OBs) [13]. Although bone alterations in IBD patients is considered to be one of the most prevalent forms of secondary OP, the underlying pathological mechanisms in IBD patients are complex and still relatively unknown. In addition to low bone mass, GI disease is also characterized by its inflammatory joint disease with features similar to psoriasis, rheumatoid arthritis, and connective tissue diseases [14]. Certain extra-intestinal manifestation on joints is related to disease activity and specific location in the gut [10]. Overall, bone and joint conditions cause severe morbidity and even mortality in patients with gut diseases such as IBD, requiring a greater understanding of the connection between these two systems. In this review, we discuss and summarize the current known knowledge underlying the mechanism of gut inflammation-mediated skeletal dysfunction in humans.

## 2. Main Risk Factors of Bone Deterioration in Patients with GI Disease

Skeletal disorders present in GI disease such as IBD are thought to result from a complex interplay of environmental and host factors.

### 2.1. Genetic Factors

A complex genetic background could be a common feature in the etiology of gut inflammation and bone loss. Several genomic studies including polymorphic analysis have been investigated to show some of the susceptibility loci overlap for both diseases, including but not limited to Interleukin (IL)-1β, *TNFSF11* (Receptor Activator of Nuclear Factor Kappa-Β Ligand, *RANKL*), *TNFRSF11B* (Osteoprotegerin, *OPG*), *IL-6*, and Vitamin-D receptor (*VDR*).

#### 2.1.1. IL-1β

IL-1β is a well characterized pro-inflammatory cytokine implicated in various inflammatory disorders. The IL-1β locus has been implicated in genetic susceptibility to IBD [15,16]. IL-1β is a well-known player in bone loss in RA patients that promotes osteoclastogenesis [17]. Furthermore, Nemetz et al. showed an increased risk of bone loss in IBD patients with IL-1B polymorphism (IL1B-511, rs16944) associated with hyper secretion of IL-1β [18]. Treatment of IBD patients diagnosed with systemic onset juvenile idiopathic arthritis with IL-1β antagonists showed well-controlled systemic juvenile idiopathic arthritis (sJIA) symptoms at time of diagnosis of IBD [19].

#### 2.1.2. RANKL/OPG

RANKL and OPG are two competing molecules critical for the process of bone remodeling. RANKL promotes osteoclastogenesis by binding to RANK and activating downstream signal cascades such as the nuclear factor kappa-B (NF-κB) pathway. OPG acts as a decoy for RANKL and can competitively interfere with its binding to RANK, thus, creating a fine tuned balance between bone resorption and formation. Altered RANKL/OPG system has been described to be associated with a lower BMD found in IBD patients. Moschen et al. paradoxically showed elevated plasma OPG in IBD patients as well as high levels of OPG released from the inflamed colonic mucosa [20]. However, a negative correlation between OPG plasma levels and spine and femoral neck BMD were found in osteoporotic IBD patients. Although plasma levels of soluble RANKL is not changed, an increased numbers of RANKL+ cells were noted in the lamina muscularis of IBD patients [20]. It is possible that in these patients, OPG levels are elevated as a compensatory mechanism in order to counteract the effects of pro-inflammatory cytokines. In addition, it is still unclear how local versus systemic effects of RANKL and OPG differ. Taranta et al. [21] showed an increased RANKL/OPG ratio and lower expression of the OC inhibitory cytokines IL-12 and IL-18 [22] in the sera of untreated celiac patients, pointing to potential peripheral blood mononuclear cell-derived impact on osteoclastogenesis. Interestingly, a recent polymorphism study found that the c.-223T allele of the OPG-encoding gene *TNFRS11B* was twice more frequent in CD patients than among controls, though the average level of *OPG* is not significantly different from controls. In UC patients, *OPG* levels were significantly lower, suggesting that low *OPG* levels may be associated with bone loss in UC, but not correlated with c.-223C>T (rs2073617) polymorphism in the *TNFRS11B* gene [23]. Furthermore, RANKL has gut-intrinsic functions by acting as a critical factor for the development of M cells [24], which are involved in antigen sampling and modulation of the gut immune response. This gut-produced RANKL may escape into systemic circulation and also have impacts on the cells of the skeletal system. Therefore, a further understanding of the roles of local and systemic RANKL/OPG on bone loss in the context of IBD is required.

#### 2.1.3. *IL-6*

IL-6 is a cytokine that has both pro-inflammatory and anti-inflammatory properties, though it is widely accepted as a pro-osteoclastogenic factor. *IL-6* expression is driven by various pathways, including the NF-κB and MAPK pathways, among others. The literature concerning genetic variations of *IL-6* expression in predisposing OP in IBD patients appears contradictory. Genetic variation analysis identified *IL-6* gene as being associated with increased risk of bone loss in IBD patients [25], especially with CD [26]. However, a previous study from the Schulte group showed a genetic response to stress in IBD patients could not support *IL-6* as a major predictor for the degree of bone disease. This coincides with several contradictory findings regarding the role of IL-6 in modulating osteoclasts and osteoblasts in bone remodeling. The role of IL-6 in gut-bone axis requires further study to resolve these conflicting findings.

#### 2.1.4. Vitamin D Receptor (*VDR*)

Vitamin D is hydrophobic nutrient critical for various body processes and skeletal health. Vitamin D can be produced endogenously in the body through the function of melanocytes or can be obtained nutritionally. Vitamin D is modified to its active form, and its binding to the intracellular VDR activates its downstream functions. A recent study from Szymczak-Tomczak et al. [27] revealed that although IBD patients did not display differences in serum 25(OH)D in comparison to control subjects, a protective effect of the *VDR* gene TaqI (rs731236, c.1057T>C) allele on BMD was seen in IBD patients and controls. Particularly, a higher femoral neck bone mass was found to be associated with tt genotypes of *VDR* gene in UC patients [27]. Another study based on North Indian postmenopausal women showed the frequency of TT genotypes was highly expressed in osteoporotic women compared to controls with normal BMD [28]. The role of VDR in the modulation of bone health is under intense debate [29,30] due to a number of cohort effects (age, district, genetic background, disease activity, etc.).

### 2.2. Microbiota

The microbiome of an individual consists of more than 1000 microbial species including bacteria and single-celled eukaryote. Homeostatically, the gut microbiota (GM) provides colonization resistance and regulates immune balance bidirectionally within the epithelial barrier and tissue environment, to protect the host from invading pathogens [31]. Perturbed balance in microbial composition has been postulated to be associated with compromised immunity within the gut, and an increased susceptibility to the development of enteropathic arthropathies, such as spondyloarthropathy and psoriatic arthritis [32,33,34]. A recent cross-sectional study displayed an increased abundance in the *Clostridiaceae* family of bacteria was shared by patients with IBD-associated arthropathy and RA, potentially due to the effects of bowel surgery history [35]. Fecal microbiota study reveals patients with spondyloarthritis (SpA) had a significant abundance of *Ruminococcus gnavus*, compared with both RA and healthy controls, correlated with a history of IBD in patients [36]. This emerging evidence suggests a potentially common microbial link for inflammatory arthritis and gut inflammation. However, it remains unknown whether enteropathic arthropathies are secondary to inflammation induced by gut dysbiosis.

### 2.3. Vitamin D Homeostasis

Among IBD patients, reduced BMD or osteopenia is associated with a lack of nutrient intake and defective nutrient utilization, mainly as a result of vitamin deficiency, insufficient calcium uptake and ultimately malabsorption due to destruction of intestinal villi. Surveys on dietary habits and nutrient intake showed lower bone density in malnourished IBD patients [37] especially when undergoing a flare. The gut, bone, and kidney are the major players regulating calcium homeostasis. The calcium homeostatic cycle is regulated systemically and locally by vitamin D and parathyroid hormone (PTH) [37]. Vitamin D is absorbed primarily in the duodenum of the small intestine through dietary consumption, as well as de novo synthesis in the skin by UV irradiation. Activity of 1, 25-dihydroxyvitamin D (Vitamin D3), the biologically active form of vitamin D, is mainly functionally mediated by the vitamin D receptor (VDR) [38]. Vitamin D increases calcium absorption in the gut, prevents calcium loss in the nephrons of the kidney, and inhibits bone resorption overall. Systemic vitamin D level is reported to be positively correlated with colonic VDR expression in healthy subjects [39]. However, IBD patients showed a reduced expression of epithelial VDR in inflamed large bowel [40,41], indicating VDR may work as an important gatekeeper maintaining mucosal barrier function. GWAS analysis of the gut microbiota identified a significant association of bacterial abundance and VDR loci [42]. Although recent studies on correlation between VDR and its relationship with IBD-associated bone loss are still in debate [27,28,29,30], targeting local expression of VDR, particularly in the epithelial barrier, might be a potential solution to manipulate vitamin D homeostasis in both IBD patients and OP subjects. Del Pinto et al. performed meta-analysis to display that UC patients possess a greater association with vitamin D deficiency than patients with CD [43]. Several studies showed pediatric and adolescent populations with IBD possess greater vitamin D deficiency than healthy subjects [43,44,45,46]. The trend towards decreased vitamin D-VDR activity likely partially contributes to the poor bone health in pediatric IBD patients [47,48], although contrasting results have been shown [49]. High risks for bone disease in IBD patients are partly due to impaired intestinal calcium absorption [37]. Since vitamin D/VDR works as crucial players in maintaining gut immune function, especially through regulating gut barrier potential [50], increased production of proinflammatory cytokines due to vitamin D deficiency may also synergistically increase prevalence of both intestinal immune response and malabsorption.

### 2.4. Medication

Anti-inflammatory and immunosuppressive drugs are commonly prescribed to IBD patients for induction and maintenance of remission during treatment. Corticosteroids, such as glucocorticoids, are the standard treatment for IBD for their rapid effect on reducing symptoms [51]. However, long term and/or high dose consumption of systemic corticosteroids results in rapid bone loss, accompanied with lower BMD and increased risk of fracture [52]. Excessive glucocorticoids consumption disrupts bone remodeling through increasing OB apoptosis, elevating RANKL/OPG levels, and stimulating osteoclastogenesis [53]. Additionally, glucocorticosteroid treatment significantly increases serum fibroblast growth factor-23 level in pediatric patients with idiopathic nephrotic syndrome [54] and receiving kidney transplant [55]. Elevated systemic fibroblast growth factor-23 level was also found to be associated with low BMD in childhood IBD [56], suggesting importance of management of steroids treatment in IBD patients. Prospective clinical trials by introduction of alternative immunosuppressive, such as azathioprine, and anti-tumor necrosis factor (TNF) agents, including infliximab, showed a superior therapeutic effect in IBD patients; however, the association of BMD, drug therapy, and the prevalence of disease activity need to be analyzed further [9].

## 3. Animal Models of Gut Inflammation-Induced Bone Loss

Recent studies characterizing bone loss in animal models of intestinal inflammation, demonstrated that gut residing immune cells as well as inflammatory mediators such as inflammatory cytokines, signaling proteins and endocrine factors, contributed to GI-disease-induced bone destruction. To better understand the interaction between bone loss and immune system in gut disease, various animal models of gut disease have been utilized. Generally, animal models of gut inflammation are mainly established using genetic modification, gut-residing microbiome modification, immune cell transfer (transferring T effector cells into immune-deficient mice), or chemical induction (e.g., TNBS, DSS). These models can then be used to study the effect on the skeletal system. A list of selected animal models indicating skeletal abnormalities seen in gut inflammation is summarized in Table 1.

### 3.1. Models Based on Genetic Modification

#### 3.1.1. *HLA-B27* Transgenic (B27-Tg) Rats

Human Leukocyte Antigen (HLA)-B27 is a major histocompatibility complex class I molecule and is highly expressed on antigen-presenting cells for pathogen recognition. HLA-B27 is notably genetic associated with Ankylosing spondylitis, a class of immune-mediated arthritis termed ‘spondyloarthritis’ (SpA) [98]. The prevalence of *HLA-B27* in SpA/IBD populations is still questionable [33], but *HLA-B27* was found to be positively associated with inflammatory back disorders [99], and asymmetrical, nonerosive peripheral arthritis (namely oligoarticular) in IBD patients [100,101,102,103]. *HLA-B27* transgenic (B27-Tg) rats develop a multisystemic inflammatory disease that display two important inflammatory features, IBD and peripheral arthritis resembling human spondyloarthritis [57]. B27-Tg rats are susceptible to both alveolar bone loss and long bone osteopenia with decreased biomechanical strength, increased bone resorption, increased RANKL/OPG mRNA ratio in bone tissue [58], and enhanced osteoclastogenesis [59]. However, this model displayed normal serum level of bone formation markers, such as OCN [60] and PINP [58], which indicates that inflammation-associated bone loss in B27-Tg rats is mainly driven by increased bone resorption. Previous studies revealed HLA-B27-associated gut and joint inflammation is prevented when transgenic rats were in a germ-free environment [104]. Recent studies from Ansalone et al. [61] demonstrated that B27-Tg rats have enhanced number of bone erosive circulating monocytes, which are highly dependent upon microbiota-regulated intestinal inflammation. These findings highlight the participation of intestinal microbiome in the pathologic relation between HLA-B27-associated gut, bone loss, and joint inflammation.

#### 3.1.2. *VDR−/−* Mice

*VDR* null mice were generated to understand the role of Vitamin D3 in the gut-bone axis. In addition to bone and intestine, VDR is widely expressed in many other tissues including endocrine tissues, skin, kidney, bone marrow, and lymphoid tissues. 1,25(OH)_2_D_3_ and VDR, and together with other active co-regulatory proteins results in maintenance of calcium homeostasis by increasing calcium absorption from the intestine and preventing bone resorption. Deficiency of VDR in mice caused rickets, osteomalacia, hypocalcemia, and hyperparathyroidism, and those symptoms were prevented when a high calcium (2%) containing diet were fed in VDR null mice, which indicates that VDR deficiency induced skeletal disorder are principally as a result of failed intestinal calcium absorption [62,63]. Further studies from Xue et al. [64], using transgenic expression of VDR in the intestine of VDR-knockout mice, showed that calcium homeostasis was directly restored, and that intestinal VDR is essential for controlling bone formation. IEC-specific VDR knockout (*Vdr*(*ΔIEpC*)) mice exhibit abnormal body size and colon structure [66], and significantly-reduced skeletal calcium levels that maintain normal levels of ionized calcium in serum. Additionally, increased compensatory 1,25(OH)_2_D_3_ levels in *Vdr*(*ΔIEpC*) mice did not rescue increased bone turnover and suppressed bone matrix mineralization, leading to osteopenia [67]. Most recent studies on the roles of VDR expressed in bone cells revealed that VDR in OBs, but not in OCs, negatively regulate bone mass, since ablation of VDR in OBs results in reduced bone resorption with a decreased expression of RANKL in OBs [105,106]. Although the role of VDR and vitamin D on the progression of IBD is unclear, VDR and vitamin D may be important for proper bone remodeling and formation of high quality bone in the setting of disease.

#### 3.1.3. *IL-10* Deficiency

IL-10 is a well-known anti-inflammatory cytokine, and decreased IL-10 levels contribute to pathological development in both IBD and inflammatory bone disease. In the study from Drezner-Pollak et al., *IL-10−/−* mice spontaneously develop colitis and exhibit decreased bone mass due to decreased bone formation, without affecting bone resorption [68]. The primary defect in bone formation in *IL-10−/−* mice was evidenced by decreased bone formation rate, serum OCN level, and mineralized nodule number in bone marrow stromal cell cultures. A following study from Ciucci et al. compared bone from *IL-10−/−* mice with colitis to *IL-10−/−* mice without colitis by histomorphometry [69]. *IL-10−/−* mice with colitis developed a significant bone defect with reduced trabecular thickness, trabecular number, and bone surface density [69]. Those findings show that osteopenia and OP found in *IL-10−/−* mice is primarily due to colitis-associated intestinal inflammation, but not IL-10 deficiency. However, it has been shown that IL-10 itself inhibits bone resorption by decreasing osteoclastogenesis [107,108]. Additional work is required to reconcile these mechanistic differences.

#### 3.1.4. *IL-2* Deficiency

IL-2, a cytokine predominantly secreted by activated T cells, is critical for the development and peripheral expansion of CD4+CD25+ regulatory T cells, promoting self-tolerance by suppressing T cell responses in vivo [109,110]. *IL-2−/−* mice spontaneously develop a systemic autoimmune inflammation characterized by colitis, hepatitis, pneumonia, hemolytic anemia, and osteopenia [71,72]. Ashcroft et al. showed that *IL-2* deficient mice developed colitis and pronounced osteopenia with reduced bone formation and increased OC number, caused by elevated level of RANKL in both serum and bone marrow mononuclear cells. IL-2 deficient mice display heavily inflamed large bowel with significantly higher systemic RANKL level at 7 weeks. The abundance of RANKL may also benefit survival and activity of intestinal dendritic cells by activating pathogenic T cells [72,111]. Transfer of CD3+ T cells from *IL-2−/−* mice into lymphocyte-deficient C57BL/6-*Rag1−/−* mice, resulted in lower femoral BMD, trabecular volume, and significantly higher number of OCs. This work suggests activated T cells, due to *IL-2* deficiency, may contribute to induction of bone loss in the setting of colitis by inducing OC formation. It emphasizes the importance of IL-2 as a therapeutic target for bone defects associated with intestinal inflammation.

#### 3.1.5. *Tnf ^ΔARE^* Mice

Tumor necrosis factor (TNF) is a main factor linking the pathology of gut-bone axis. This was demonstrated by TNF transgenic mouse model (*Tnf ^ΔARE^* mice), generated by deleting TNF AU-rich elements (ARE) responsible for destabilizing activity on the TNF message. These mice spontaneously develop chronic inflammatory polyarthritis at weeks 5–6 and Crohn’s-like IBD at week 6 [73]. Moreover, these mice showed enhanced circulating TNF levels and profound capacity of macrophages and T cells to produce TNF. This model is effective for studying severe systemic inflammatory diseases as seen in clinical human patients, since most patients have systemic inflammation that is not isolated to a single organ system. However, the systemic nature of this model makes it difficult to understand the specific effect of gut inflammation on other regions of the body.

#### 3.1.6. *gp130^ΔSTAT/ΔSTAT^* and *gp130^Y757F/Y757F^* Mice

The proinflammatory cytokine IL-6 and its receptor gp130 are essential for the pathogenic development of both intestinal and bone disease, such as RA. Genetic manipulation of the IL-6/leukemia inhibitory factor (LIF) cytokine axis through sustained activation of gp130 in mice (*gp130-KI mice/gp130^ΔSTAT/ΔSTAT^*), causes severe joint disease with features representative of RA, and interestingly, GI ulceration [74]. Those mice showed a severe chronic synovitis, degraded articular cartilage associated with disrupted chondrocyte differentiation, but normal trabecular bone volume (BV/TV) and turnover [75]. *gp130^ΔSTAT/ΔSTAT^* mice display sustained gp130-dependent signal transducer and activator of transcription (STAT) signaling cascades due to impaired STAT-mediated induction of suppressor of cytokine signaling-1 (SOCS-1), which normally functions to limit gp130 signaling [112]. Dysregulation of STAT1 and/or STAT3 activity is associated with human hematological and intestinal disease [113], such as gastric cancer [114]. Unlike *gp130^ΔSTAT/ΔSTAT^* mice, BV/TV was reduced in gp130 receptor mutant mice (*gp130^Y757F/Y757F^*) due to highly activated bone turnover [75]. The work from this model, combined with evidence that IL-6 levels are elevated in IBD patients, suggests a role for gut-induced IL-6 expression as a mediator of bone quality.

#### 3.1.7. Hematopoietic Cell-Specific *STAT3*-KO Mice (STAT3-CFF; *Tie2^Cre+^-Stat3^fl/fl^*)

Signal transducer and activator of transcription 3 (STAT3) is a transcriptional mediator responsible for the expression of many inflammatory cytokines, including IL-6, IL-10, and G-CSF [115], exerting its role in a cell-specific way [116]. Mice with hematopoietic cell-specific disruption of *STAT3* gene (STAT3-CFF) generate an osteoporotic phenotype with hyperproliferated myeloid lineages and increased number of OC precursors, indicating existence of higher osteoclastogenic factors in those mutant mice [117]. Another study from Welte et al. showed STAT3-CFF mice displayed a CD-like pathogenesis in both the small and large intestine, characterized with segmental inflammatory cell infiltration, ulceration, and formation of granuloma [76]. An alternative study also documented that STAT3-CFF mice have defective stem/progenitor cells with mitochondrial dysfunction, increased reactive oxygen species (ROS) production, and a rapid aging-phenotype [77]. All these findings suggest that STAT3 signaling pathway and its associated factors, such as ROS and mitochondrial function, contribute to both inflammatory bowel and bone diseases. Work remains to be done on connecting the two organs in this model and understanding how bowel-specific STAT3 signaling may affect bone remodeling.

#### 3.1.8. *A20* (Tumor Necrosis Factor α-Induced Protein3; *TNFAIP3*) Deficiency

Nuclear factor-κB, a key nuclear transcriptional factor, is a central player in regulating expression of a variety of genes encoding proinflammatory cytokines, adhesion molecules, chemokines, growth factors, and inducible enzymes [118,119]. NF-κB itself is activated by many different stimuli ranging from inflammatory cytokines to genotoxic stress. Pathogenesis in both IBD and inflammatory bone disease is characterized by elevated inflammatory cytokines, many of which are regulated through activation of NF-κB signaling. Dysregulation of NF-κB inhibitory pathways in mice can cause severe systemic inflammation. One such model is characterized by reduced *A20* (tumor necrosis factor α-induced protein3; *TNFAIP3*) expression (*A20*-KO mice), which normally regulates NF-κB activation. *A20* deletion triggers severe inflammatory conditions in multiple organs, including livers, kidney, intestines, joints, and bone marrow, and dies prematurely [120]. Studies on cell-specific A20 function, by using conditional knockout mice, showed that A20 deficiency in myeloid cells (*A20^myel^-KO*) results in spontaneous development of severe destructive polyarthritis resembling RA [78]. Although intestinal epithelial cell (IEC) specific A20 knockout mice, and *A20^myel^-KO* mice showed preserved intestinal integrity, models combining IEC and myeloid A20 deletion display ileitis and severe colitis, characterized by elevated inflammatory cytokine pool [79]. This work suggests that both an extracellular source of inflammatory stimulation, combined with defective intestine-specific intracellular signaling, may be required for the development of gut disease.

#### 3.1.9. IKK2ca^IEC^ Mice

To further interrogate the role of NF-kB signaling in the gut-bone axis, our lab generated an IEC-specific model of NF-κB overactivation. The findings from our studies using IEC-specific constitutively activated IKK2 mice (IKK2ca^IEC^) model highlight direct and crucial role of intestinal NF-κB signaling in linking chronic gut inflammation with bone loss [80]. IKK2ca^IEC^ develop a mild inflammation in small intestine [81]; however, with significantly elevated serum and epithelial level of inflammatory cytokines. Importantly, IKK2ca^IEC^ mice recapitulate the majority of the phenotypes observed in chemically (Dextran sulfate sodium/DSS) induced colitis, such as altered gut-residing cells and elevated osteoclastogenic cytokines. Furthermore, moderate attenuation of NF-κB signaling by using conditional deletion of one allele of IKK2 in IECs or pharmacological inhibition of IKK2, partially attenuated circulating levels of inflammatory cytokines, such as IL-17, and halted colitis-associated bone loss. This indicates that NF-κB signaling in intestinal cells is not only critical for gut inflammation-mediated damage, but also extra-intestinal effects.

#### 3.1.10. *FXR−/−* Mice

The farnesoid X receptor (FXR), a nuclear receptor for bile acids, is highly expressed in intestine and liver and responsible for maintaining bile acid homeostasis. FXR displays various functions in different organs, and the FXR antagonists have been developed for pre-clinical and clinical application for management of liver and other metabolic disease [121]. *FXR−/−* mice display compromised intestinal epithelial barrier due to increased bacterial invasion [122], increased colon cell proliferation and apoptotic goblet cells, accompanied with upregulation of genes involved in cell cycle progression and inflammation, such as cyclin D1 and IL-6 [83]. The changes in the intestinal microbiota during metabolic dysfunction, such as obesity and alcoholic liver disease, have been shown to be associated with FXR [123,124]. FXR deficiency leads to cell proliferation, inflammation, and tumorigenesis in gut, and FXR can protect small intestine from bacterial invasion and colonization [125]. Cho et al. demonstrated that *FXR−/−* mice display reduced bone formation rate as well as low trabecular bone volume and reduced thickness [84]. Evaluation of FXR effect on bone cell differentiation showed that mice with FXR deficiency had reduced OB differentiation [84], and enhanced OC generation [85]. Zheng et al. [85] assessed role of FXR in different pathological bone loss models (calvarial injection of LPS, in vivo rosiglitazone treatment, ovariectomy (OVX) surgery, and unloading-induced bone loss), and reported that deficiency of FXR accelerated bone loss in animal model of postmenopausal OP and unloading-induced bone loss. Taken together, these findings strengthen both biological and pathological function of FXR in bone loss and highlight a possible role of FXR in connecting intestinal inflammation and its-related bone disease when FXR is downregulated. Further work has to be performed to determine how intestinal inflammation may modulate FXR ligands to signal to the skeletal system through cell specific deletion models.

#### 3.1.11. *Mdr2−/−* Mice

Multi-drug resistance 2 (encoded by *mdr2/Abcb4*), known as a canalicular phospholipid flippase, is responsible for transport of phospholipid into bile by increasing efflux of intracellular hydrophobic drugs observed in multidrug resistance (MDR) [126,127]. The *mdr2−/−* mouse spontaneously develops severe biliary fibrosis due to an accumulation of toxic bile acids, and is considered as a model of primary sclerosing cholangitis (PSC) [128]. PSC is a cholestatic liver disease characterized by chronic inflammation and progressive destruction of the bile ducts. PSC displays remarkable associations with IBD, dysbiosis and higher incidence of fracture risk [129,130]. *Mdr2−/−* mice display an altered GM, intestinal barrier dysfunction, and activation of NLRP3 inflammasome within the gut–liver axis. Interestingly, healthy control mice transferred with *Mdr2−/−* microbiota develop significant liver injury, highlighting the key function of intestinal dysbiosis in the progression of liver disease [86]. Schmidt et al. demonstrated decreased bone mass in PSC patients is associated with increased bone resorption, as well as increased peripheral blood T helper (Th)17 cells. Experimental studies showed that *Mdr2−/−* mice had a higher Th17 cell frequency in liver. Analysis of skeletal phenotype displayed that absence of Mdr2 in mice resulted in osteopenia with reduced trabecular bone mass, and this phenotype was prevented by the additional *Il17af* deficiency [87]. An increased Th17 response to microbial stimulation was found in PMC patients with UC [131]. These findings point out that mdr2-regulated dysbiosis may contribute to two major complications found in PSC patients: IBD and bone loss, potentially through modulation of immune cell populations.

### 3.2. Models Based on Dysbiosis-Associated Gut Inflammation

The gut is an especially unique organ system due to the present of its vast microbiome that is critical for healthy function but also pathophysiology. Dysbiosis is highly associated with GI disease, characterized by imbalanced microbiota population as well as microbial metabolism. Therefore, recent work using animal models to manipulate host microbiota offer a method to understand the role of the microbiome in gut-related disease and associated bone loss. Animal studies to explore dysbiosis-regulated bone loss were mainly performed by using germ-free (GF) animals and antibiotic treatment [132].

#### 3.2.1. Germ-Free (GF) Models

GF models are established by raising animals in sterile conditions, which result in a lack of microbiota in those animals and an immature mucosal immune system [133]. This allows for colonization with specific microbiota to determine the effect of individual bacterial species on physiology. GF animals display enlarged ceca, reduced total mass of intestine and total surface of small intestine, shorter crypts of small intestine, and decreased cellularity in lamina propria, compared to conventionally raised animals (CONV-R) [134]. GF mice displayed increased trabecular bone mass due to compromised osteoclastogenesis, compared with CONV-R mice. Colonization of young GF mice (3 weeks old) with normal GM (for 4 weeks) normalized bone mass to a level similar to CONV-R mice [88]. Subsequent study from Yan et al. showed long-term colonization of 2 month old GF mice with conventional specific pathogen-free GM for 8 months results in equalization of bone mass between GF and conventional groups. Long term conventionalization of GF mice also increased longitudinal and radial bone growth, although an acutely reduced bone mass was found when a short-term (1 months) colonization was conducted [90]. Colonized GF mice also showed an increased circulating level of insulin-like growth factor 1 (IGF-1), a hormone known to effectively regulate skeletal growth. Together, these observations suggest a complex relationship between GM and bone remodeling over the lifetime of the animal. Hence, the role of GM and their impact on the skeletal system requires further work developing clinically-relevant strict GF animal quality control procedures to tease apart these effects.

#### 3.2.2. Antibiotics Treatment Models

An alternate method to study GM is by using antibiotic treatment to kill specific subsets of bacteria. This method has the benefit of being highly translational in order to study the impact of antibiotics in humans as well. Treatment with broad-spectrum antibiotics commonly reduce or deplete microbiome population in gut, with increased size of cecum, decreased proliferative cells in small intestine and colon, but with fewer effect on overall colon length [132]. Yan et al. showed antibiotic treatment of CONV-R mice decreases serum IGF-1 level and inhibits bone formation [90]. However, supplemental treatment with short-chain fatty acid (SCFAs), one of the products of microbiota metabolism, restored serum IGF-1 and bone mass level in antibiotic-treated CONV-R mice [90], suggesting that metabolic products of bacteria can regulate the host skeleton. The other alternative study showed a mild range of antibiotic treatment at early life significantly increase BMD in CONV-R mice [135], as well as serum glucose-dependent insulinotropic polypeptide (GIP), an incretin hormone promoting bone formation [136]. Those results are consistent with the finding that daily injection of GIP could rescue bone loss found in OVX animal model [137]. Importantly, multiple studies provide evidence that supplementation of probiotics (i.e., *Lactobacillus reuteri*, *Lactobacillus rhamnosus GG* (*LGG*), *Lactobacillus paracasei* and/or *Lactobacillus plantarum*) in OVX mice can limit bone loss by dampening osteoclastogenic cytokine production in bone marrow and/or intestine due to ovary malfunction [138,139,140]. Based on experimental data, intestinal microbiota display a great potency to regulate bone metabolism via regulating host metabolism, immune response, and endocrine factors. Normalization of the intestinal microbiota directly or indirectly, for example, dietary interventions by using pro-and/or synbiotics, may be provided for pre-clinical application for treating metabolic disease and GI disorder-associated bone disease.

### 3.3. Chemical Irritant-Induced Model

The use of chemical irritants to induce IBD in mice has been one of the most highly utilized, non-genetic models for studying the impact of IBD systemically. Several such models have been developed, and although they do not mimic the direct pathophysiology of human IBD, they display similar phenotypic changes as human patients [141].

#### 3.3.1. Trinitrobenzene Sulfonic Acid (TNBS)

Colitis induced by trinitrobenzene sulfonic acid (TNBS), a classical haptenating agent, is mainly conducted through intrarectal administration to trigger production of immunogenic proteins in the colon, thereby initiating a mucosal immune response such as increased cytokine production [142]. An early report by using TNBS-administration in rats showed that significant cancellous bone loss (33%) could be observed 3 weeks after treatment, due to reduced cancellous bone formation rate (<30% versus control). Following the resolution of the inflammatory process, TNBS-treated animals exhibit increased bone formation by 12 weeks into the experiments [143], indicating the potential for rescue after disease resolution. A recent study by Metzger et al. reported systemic inflammation developed in TNBS-induced IBD elevates proinflammatory cytokines levels (TNF-α, IL-6, RANKL) and decreases sclerostin, a known bone formation inhibitor, in osteocytes [91]. These findings highlight potential contribution of bone formation modulating cells, OB, and osteocytes, in bone loss associated with TNBS-induced IBD. Other studies demonstrated increased TNF and interferon-gamma (IFNγ) in TNBS-treated mice can downregulate expression and activity of Klotho, an anti-inflammatory protein by supporting renal Ca^2+^ homoeostasis [70,92].

#### 3.3.2. Dextran Sulfate Sodium (DSS)

An alternate chemically induced IBD model is the well-established DSS colitis model. This model utilizes DSS to induce dysfunction of intestinal epithelial barrier, thereby resulting in exaggerated entry of luminal bacteria or antigen into mucosa [144,145]. The effect of DSS on the induction of bone loss is dependent upon DSS dosage, treatment periods, and animal background (age, sex, and species). Low dose treatment of DSS (1%) in 5–6 weeks old male mice for 15 days induced a moderate intestinal inflammation without weight loss [95]. Those mice displayed significant trabecular bone loss with increased TNF-α in bone and colon, which is negatively correlated with OB activity, leading to decreased bone formation rate, but few effects on cortical bone. Interestingly, low dose DSS treatment induces a redistribution of fat storage from subcutaneous to visceral sites, which is reported to be more proinflammatory and consistent with disease pathologies (e.g., Creeping fat accumulation) found in CD patients [146,147,148]. Studies from Harris et al. reported a transition in bone composition occurred following active IBD phases [93]. During active disease phase, DSS-treated mice showed a significant decrease in tibial trabecular and cortical bone parameters, lower vertebral trabecular bone parameters, but few effects on calvaria BMD. Active inflammation in intestine is associated with dramatically increased serum level of TNF-α and reduced IGF-1, associated with decreased bone length, growth plate thickness, and cartilage markers. However, both OC and OB activity declined during active disease stage, indicating decreased overall bone remodeling. During disease recovery, DSS-treated mice recovered body weight and skeletal response. Different from previous groups, Hamdani et al. [149] used Balb/C mice to investigate DSS colitis-induced bone loss, and they found reduced femoral bone mass resulting from suppressed bone formation and increased bone resorption in the DSS-colitis model. Interestingly, endogenous bone marrow mesenchymal stem cells (BMMSCs) have an immunosuppressive capacity, maintained by autophagy, that can improve the intestinal pathologic index in DSS-colitis mice, highlighting reciprocal interaction between the gut and bone [150]. However, BMMSCs from OVX mice lost this immunoregulatory capacity, exhibited decreased osteogenic differentiation and increased adipogenic differentiation. The restoration of autophagy by rapamycin rescued BMMSCs’ healing capacity and attenuated the OP phenotype in OVX mice. These different findings indicate the despite its simplicity and wide application, DSS-induced colitis is complicated by various factors, which make it challenging to study the impact of colitis on bone. However, this can also provide insight into the various modes and severity of intestinal inflammation-associated bone loss observed in human patients.

### 3.4. Immune Cell-Transfer Induced Model (CD45RB Model)

A prominent feature of chronic gut inflammation in IBD is the presence of activated inflammatory T cells as well as their secreted cytokines. Differentiation/expansion of donor (Wild-type)-derived naïve (CD4+CD45RB^Hi^) T cells in immune-deficient recipient (Rag−/−; *scid/scid* mice) induces transmural colitis with severe inflammation within small bowel [151]. Experimental data showed that, in addition to colitis, *scid/scid* mice transferred with CD4+CD45RB^Hi^ T cells (CD45RB mice) developed osteopenia characterized with decreased total (trabecular and cortical) BMD, increased OC number in long bone surface and decreased number of OBs [96]. CD45RB mice also displayed bone marrow inflammatory cells expressing higher TNF-α. Prophylactic treatment with Fc-OPG, a soluble decoy receptor binding and neutralizing RANKL, increased bone density in CD45RB mice by preventing OC function, normalizing serum Ca^2+^ and Pi concentration, but failed to recover gut inflammation. A similar therapeutic function of OPG in gut-inflammation associated bone loss was also found in IL-2−/− colitis mice, in which both skeletal abnormalities and colitis score were reduced by modulation of RANKL-RANK interactions with exogenous administration of Fc-OPG [72]. The potential critical effect of systemic and intestinal RANKL in bone loss and osteoclastogenesis is also revealed in our recent study [80]. To sum up, the role of the RANK/RANKL signaling pathway in location inflammation is still needed to be addressed in different gut inflammation models.

## 4. Networks of Gut-Residing Factors Regulating Bone Marrow Microenvironment and Bone Loss

The intestine and bone marrow are two representative central reservoirs of immune cells. Inflamed gut environment, driven by unrestrained immune-cell activation, increased pro-inflammatory cytokine production, associated with altered resident microbiotas and dysfunction of epithelial barrier, results in increased inflammation in bone marrow microenvironments and subsequently decreased bone mass (Figure 1).

### 4.1. Intestinal Barrier-Regulated Bone Loss

Intestinal epithelial barrier, an integral component of innate immunity, provides a shield against microorganisms harbored in lumen. The intestinal barrier is mainly composed of a single layer of various cell types (i.e., IEC, enterocytes, goblet cells, neuroendocrine cells, tuft cells, Paneth cells, and M cells), which respond to various extracellular stimuli and provide priming and activation signals to a diversity of immune cells residing in the lamina propria to promote an effective inflammatory response [152,153]. The intestinal epithelial cells make up the majority of the luminal surface where they are connected by tight junctions (TJ), which are responsible for connecting adjacent epithelial cells and are critical for controlling interepithelial permeability and selectively controlling passage of small molecules. Studies in active CD patients showed increased intestinal paracellular permeability which is associated with altered TJ protein complex, such as suppressed expression of transmembrane protein occludins, and claudins 5 and 8, but upregulated claudin-2, a mediator of leaky gut barrier [154]. These emphasize the key role of TJ protein in shaping epithelial structure, restraining pathogen passage, and maintaining gut barrier function.

Current knowledge of the crosstalk between intestine and bone during gut inflammation is based off both clinical and animal models [155]. Pediatric IBD patients showed an increased likelihood to develop OP or osteopenia due to impaired barrier function. Early studies showed that premature birth can increase intestinal permeability and, surprisingly, bone mineral content [156]. Another study found that infants with low birth weight were more likely to develop bone metabolic disease [157]. The effects of intestinal health during early development can clearly have long-term effects on bone health, especially since younger children undergo higher rates of bone remodeling. This could also be potentially associated with high permeability of intestinal barrier during early phase of development. Breastfeeding may allow passage of multiple components, including lactoferrin, transforming growth factor beta (TGFβ), and epidermal growth factor (EGF) from the mother’s colostrum to the infants’ blood. Those components improve infants’ intestinal permeability by promoting TJ protein expression [158], as well as provide easily digestible nutrition for infants. In addition, IgA antibodies found in breast milk have been shown to improve gut microbiota and overall intestinal health, reducing the risk of IBD. Although more evidence is needed to prove an association between bone loss and intestinal barrier dysfunction, studies on animal models using chemically-induced barrier permeability [93,149] and by targeting epithelial function through genetic modification [80], support this notion. A major drawback of this clinical finding is small sample sizes and highly specific patient populations that may not represent broad patient groups. Typically, proinflammatory cytokines TNFα and IL-1β, which are increased in IBD patients, have been shown to directly increase IEC permeability by interrupting TJ proteins, associated with increased activation of NF-κB signaling, the primary inflammatory response pathway [159,160,161,162]. Those results are consistent with our findings in IKK2ca^IEC^ mice, wherein constitutive activation of NF-κB signaling pathway in IECs [80], which mimics chronic inflammation, can induce gut damage and directly affect bone regulating cells (such as OC precursors) via gut-secreted cytokines.

A recent update shows that intestinal epithelial barrier can respond to endoplasmic reticulum (ER) stress under inflammatory conditions, and Genome-wide association studies, (GWAS) based on a large cohorts of IBD patients undercovering over 300 susceptibility genes, revealed a high correlation between endoplasmic reticulum stress as well as unfolded proteins (UPR)-related genes and the pathogenesis of IBD [163,164]. Reports on animal studies showed that mice with epithelial-specific deletion of X-box-binding protein (XBP1) spontaneously developed enteritis, exhibited a higher susceptibility to induce colitis due to dysfunction of Paneth cells and high sensitivity to bacterial products (flagellin) and TNFα [165]. Another recent study emphasizing the importance of ER protein in epithelial function displayed mice with epithelial specific disruption of Inositol Requiring Enzyme1α, an ER transmembrane protein serving as the major sensor under stress, spontaneously developed colitis with a loss of goblet cells and failure of intestinal epithelial barrier function [166]. The importance of ER stress signaling is not only reflected by its role in maintaining physiology, but also by its high response to genetic mutations, oxidative stress, process of aging, and/or various environmental factors that can lead to different diseases such as obesity, inflammation, diabetes, GI disorders, and rheumatic disease (RA) [167,168,169]. In spite of the lack of evidence proving direct link between ER stress presented in IBD and bone disease, the promising effect of UPR in bone cells differentiation and functions has been investigated in multiple studies. Scheiber et al. showed that G610C osteogenesis imperfecta mice had longitudinal bone growth retardation, characterized by accumulated hypertrophic chondrocytes expressing ER dilation, and mutated type I collagen which induce osteoblast dysfunction due to ER stress [170]. ER stress also reflects one of the promising therapeutic mechanisms during rheumatoid disease process. For example, induction of UPR signaling by inflammatory cytokines and autoantibodies can act as upstream pathway to induce inflammatory responses during RA pathogenesis [169]. Upregulated expression of representative ER stress markers, including GRP78, IRE1, XBP1s, ATF6, and eIF2α-P, were found in macrophages and synovial tissues from RA patients [171,172]. Tauroursodeoxycholic acid (TUDCA), an endogenous chemical chaperone that protects cells against ER stress, has been shown to attenuate intestinal inflammation and barrier disruption in various disease models, such as DSS-induced colitis [173,174] and non-alcoholic fatty liver disease [175]. The effect of TUDCA in bone was also investigated in experimental models which displayed that TUDCA was comparable to recombinant human bone morphogenetic protein-2 possessing osteogenic potential in a mouse spinal injury model [176], and an alternative treatment to restore OA cartilage by balancing intracellular cholesterol levels in chondrocytes [177]. Therefore, targeting dysregulated UPR signaling in gut and/or bone marrow microenvironment may be proposed as new therapeutic strategy for IBD-associated bone destruction.

Another recent potential theory that links intestinal barrier dysfunction and skeletal failure presented in IBD could be autophagy, a conserved catabolic process by which cells control degrading damaged protein, dysfunctional organelles and clearance of bacteria [178]. Activation of autophagy machinery has been indicated in various human diseases [179]. GWAS analysis identified autophagy protein (ATG16L1) as an IBD susceptibility gene [180], representing key roles in both human and mouse intestinal Paneth cells [181]. Adolph et al. reported that mice with epithelial specific compromised UPR (XBP1) and autophagy function (ATG16L1) spontaneously developed severe CD-like transmural ileitis [182], accompanied with NF-κB overactivation-mediated intestinal inflammation. Several studies revealed regulation of autophagy on bone cell development and function, as well as the effects of autophagy on bone microenvironment are context-dependent [183,184,185,186,187]. Wu et al. reported that IEC-specific VDR regulated ATG16L1 level at both transcriptional and translational level [188]. Additionally, the negative correlation of intestinal VDR expression and ATG16L1 is found in UC patients as well as IL-10−/− mice with colitis. Bone marrow-derived macrophages from IL-10−/− mice displayed an impaired induction of autophagic pathway upon LPS stimulation [189], implicating selective local activation of autophagy, i.e., in gut or bone, may explain osteopenia shown in IL-10−/− mice [68]. Qi et al. presented that autophagy maintains the immunoregulatory capacities of BMMSCs in modulating T cells apoptosis, gut inflammation in DSS-colitis mice, but BMMSCs from OVX/OP models failed to exert regeneration capacity [150]. Clinical data also showed promising therapeutic effect of mesenchymal stem cells for the treatment of UC [190]. Overall, these findings indicate that modulation of novel signaling pathways, such as NF-κB, UPR, and autophagy, can regulate cytokine production in the inflamed gut environment to limit bone destruction.

### 4.2. Effect of Gut-Derived Cytokines on Bone Marrow Microenvironment

Patients with IBD display elevated levels of circulating pro-inflammatory cytokines with osteoclastogenic function, such as TNFα, IL-1β, IL-6, IL-11, IL-17, and prostaglandin E2 [7]. These osteoclastogenic factors presumably act on OC precursors residing in bone microenvironment to promote OC differentiation and bone resorbing capacity. Additionally, some of those cytokines disrupt bone formation by inhibiting OB differentiation and bone formation. Thus, the dual effects of these inflammatory cytokines lead to a substantial bone erosion effect and further result in skeletal anomaly. Neutralization of pro-inflammatory cytokines, such as TNFα and IL-6, which are increased in the circulation of active IBD patients, can therapeutically recover IBD-associated bone loss [191], highlighting the important role of effector cytokines in the effects of IBD on bone remodeling. In murine studies, varying contributions of cytokines in IBD-associated bone defects have been reported.

At the onset of IBD, dysfunction of epithelial barrier with higher permeability cause accumulation of GM and increased innate immune cells response, excessive influx and activation of lymphocytes including neutrophils, macrophages, dendritic cells, and innate lymphoid cells (ILC) [192]. These cells produce high quantities of pro-inflammatory cytokines (IL-1β and TNFα), which are more actively produced at the onset of gut inflammation [193,194]. Our most recent studies showed both IEC and ILCs, the representative innate immunity regulating components, are responsible for initiation of murine osteopenia due to intestinal barrier dysfunction [80]. Furthermore, since IBD is characterized by recurrent chronic inflammation, the adaptive immune response plays a key role in the development of IBD-associated bone loss [195]. Gut inflammation has been characterized by overactivated pro-inflammatory cytokine-producing T-helper cells, such as Th1 and Th17, in addition to ineffective anti-inflammatory regulatory T cells to resolve the inflammation [195]. Unrestrained gut-immunity in IBD is associated with divergent responses in lymphoid and myeloid hematopoiesis in the bone marrow. Trottier et al. and others showed that colitis in mice induces significant increase of neutrophils, monocytes, and granulocytic lineage, but a decline of lymphocyte population (B and T cell lineage) in bone marrow and peripheral blood, which is consistent with the highly infiltrated monocytes and neutrophils in intestine [196,197]. However, the key drivers of inflammation in the IBD-derived bone marrow still remain to be addressed. Ciucci et al. identified in two characterized mouse models of IBD (IL-10−/− and CD45RB model), as well as CD patients, that bone marrow-Th17 cells expressed higher levels of osteoclatogenic cytokines, including IL-17 and TNFα upon gut inflammation [69]. These findings are consistent with previous observations that intestinal inflammation in IL-2−/− mice is associated with activated T cells accumulation in the bone marrow and highly produced RANKL [72]. Oostlander et al. also suggested an important role of IL-17 in OC differentiation in CD patients [198]. Attenuation of NF-κB signaling during gut inflammation in mice limit bone loss by modulating circulating IL-17 level [80], highlighting the key function of IL-17 in gut-bone inflammation.

To date, direct evidence supporting the effect of gut-residing cells as well as their secreted cytokines on bone remodeling still needs further exploration. However, gut microbial dysbiosis may be the key driver to aberrant immune response, often accompanied with imbalanced production of inflammatory cytokines. Data of the Human Functional Genomics Project, derived from in vitro stimulation experiments, demonstrate that TNFα and IFNγ production are more directly influenced by the GM [199], suggesting the possibility of modulating the host immune response by manipulating microbiota-mediated factors, i.e., cytokines, rather than direct targeting of the immune cells. Mice in GF condition exhibit increased bone mass and compromised marrow CD4+ T cells and osteoclastogenic cytokines, which can be normalized by conventionalization [88,89,139,200]. Shaping of GM in B27-tg rats modulates onset of gut and joint inflammation [104]. Several experimental studies also emphasize the critical role of microbial dysbiosis in linking metabolic dysfunction-associated gut inflammation and bone loss [201]. For example, mice with ovary dysfunction display an increased intestinal permeability [139,202] with altered intestinal TJ and cytokine gene expression [202]. Probiotics supplementation improves bone loss and intestinal permeability following OVX [138,139,140]. GF mice are protected from bone loss and barrier dysfunction due to sex-steroid deprivation, by dampening bone marrow and gut-derived proinflammatory cytokines [139]. Therefore, inflammatory cytokines and GM can be seen as the key drivers regulating inflamed gut and it manifested bone disorder, which can be a potential clinical frontier to limit skeletal disease activity.

## 5. Closing Remarks

During the past decade, plenty of research has focused on the skeletal abnormalities seen in IBD patients or animal models. Many risk factors including genetics and environmental elements are suggested to be associated with reduced bone mass in IBD patients. The various pathological mechanisms by which gut inflammation can exacerbate deterioration bone architectures are still under on-going debate. Development of a specific therapeutic strategy coping with a number of cohort effects (age, district, genetic background, disease activity, etc.) need to be considered by physicians and researchers [203,204]. Progress on experimental studies of animal models implicates dysbiosis of the intestinal microbiota, and its related inflammatory cytokines, is the key player driving systemic inflammation and subsequently resulting in bone loss. However, the understanding of pathogenic mechanisms defining interactions between bone and intestine is still at an early stage. More clinical and preclinical studies with well-designed human and animal models of IBD need to focus on interrogating major inflammatory mediators and key players that can influence gut-bone axis signaling in order to reduce the morbidity associated with gut-disease mediated bone loss.

## Figures and Tables

**Figure 1 ijms-20-06323-f001:**
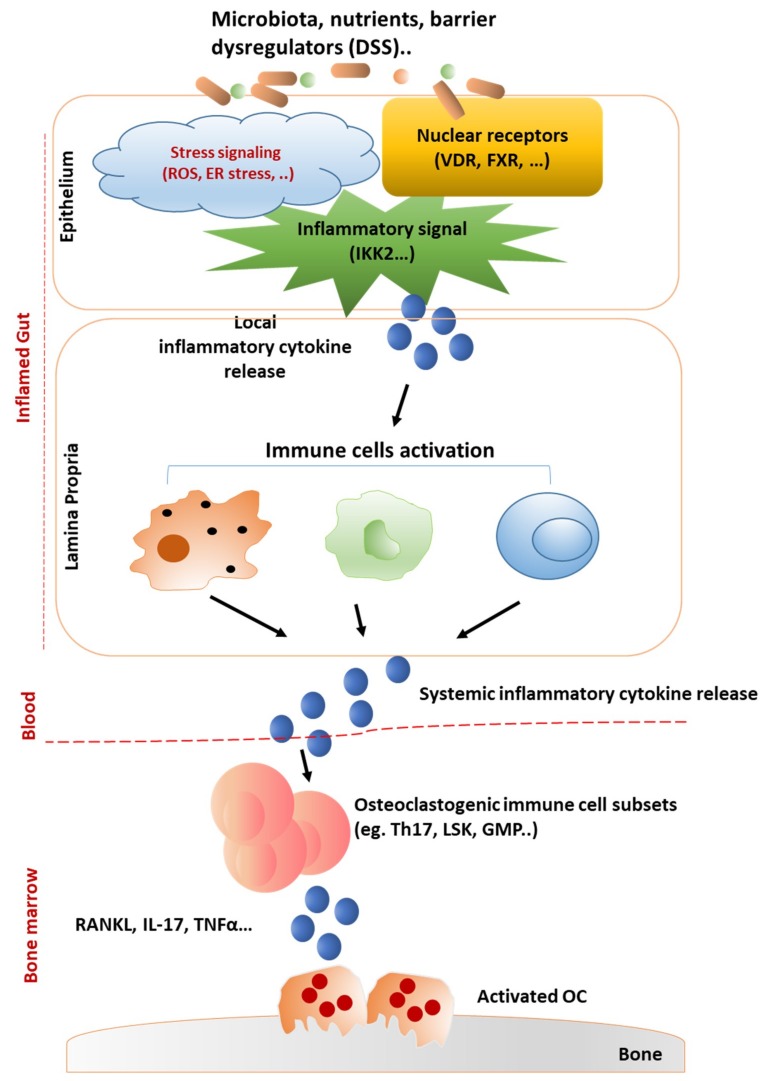
Networks of gut-immune response in bone loss. Many factors, including microbiota and/or barrier-damaging factors (e.g., DSS), have been validated in triggering gut inflammation, such as IBD. In inflamed gut, epithelial barrier function can be regulated by nuclear receptros (VDR, FXR), inflammatory signals (NF-κB), and various stress stimulators (ROS, Endoplasmic Reticulum (ER) stress). Various inflammatory responses can be activated in the lamina propria by the infiltrating bacteria and/or local released inflammatory cytokines from epithelium cells. Dysregulation of the balance between mucosal immune cells orchestrate cytokines production and releasing to blood stream. Systemically provided inflammatory cytokines could be transmitted to bone marrow to enable activation of osteoclastogenic immune cells and cytokines, leading to activated bone resorption process by osteoclasts (OC).

**Table 1 ijms-20-06323-t001:** Animal models indicating linkage between gut-bone axis signaling.

**Models Based on Genetic Modification**				
**Animal Strains**	**GI Phenotype**	**Skeletal Phenotype**	**Systemic and Other Organ Phenotype**	**Reference**
*HLA-B27 Tg* rats	IBD dysbiosis	Spondylarthritis; RANKL/OPG ↑ osteoclastogenesis ↑	circulating monocytes ↑ serum OCN, PINP: n.s.	[57,58,59,60,61]
*VDR−/−* mice	normal intestine; colonic 8-OHdG ↑ Ca^2+^ absorption ↑	rickets, osteomalacia; femur BMD and MAR ↓	BW↓ Serum OCN, 25-OH-D_3_, Ca^2+^ ↓ 1,25(OH)_2_D_3_ and PTH ↑	[62,63,64,65]
*Vdr*(*ΔIEpC*) mice	intestinal Ca^2+^ absorption ↓	osteopenia; bone turnover ↑; BV/TV, Ct.Th., Tb.N., Tb.Th. ↓; mineralization and skeletal Ca^2+^ ↓; RANKL/OPG ↑	Serum PTH, 1,25(OH)_2_D_3_, CTX, and OCN ↑	[66,67]
*IL-10−/−* mice	colitis	bone mass, Tb.Th. and Tb.N. ↓ bone formation ↓ %Th17-TNFα+ in BM ↑	serum OCN ↓ renal Klotho ↓	[68,69,70]
*IL-2−/−* mice	colitis; colonic dendritic cells, macrophages, and antigen-presenting CD4+T cells ↑	osteopenia; bone formation ↓, OC.N.↑, BM monocytes ↑, RANKL, OPG ↑	serum RANKL, OPG, MCP-1, IL-6, TNFα, and interferon-gamma (IFNγ) ↑	[71,72]
*Tnf^ΔARE^* mice	CD-like IBD	spontaneous polyarthritis	serum TNF ↑	[73]
*gp130 ^ΔSTAT/ΔSTAT^* mice	ulceration	chronic synovitis; degraded articular cartilage; chondrocyte differentiation ↓ trabecular BV/TV n.s.	N/A	[74,75]
*gp130 ^Y757F/Y757F^* mice	gastric cancer	BV/TV ↓ bone turnover ↑	N/A	[75]
*STAT3-CFF* (*Tie2^Cre+^-Stat3^fl/fl^*) mice	CD-like IBD, immune cell infiltration TNFα ↑, INFγ ↑	BV, Tb. Th., and Tb.N. ↓ OC.N. ↑ mitochondrial dysfunction in stem/progenitor cells and ROS ↑	N/A	[76,77]
*A20^myel-KO^* mice	less stable microbiota	severe polyarthritis macrophage NF-κB activity and TNF ↑ osteoclastogenesis ↑	serum inflammatory cytokines ↑ Spleen and inguinal lymph nodes Th17 ↑	[78,79]
*IKK2ca^IEC^*	mild gut inflammation at 8–10 weeks, inflammatory cytokines in IECs and ILC3 ↑	BV/TV ↓ Osteoclastogenesis ↑	serum inflammatory cytokines and CTX ↑	[80,81,82]
*FXR−/−* mice	barrier dysfunction apoptotic goblet cells	BFR, Tb. BV. and Tb. Th. ↓ OB differentiation ↓ osteoclast generation ↑	N/A	[83,84,85]
*mdr2−/−* mice	barrier dysfunction; dysbiosis	osteopenia trabecular bone mass ↓	N/A	[86,87]
**Models Based on Dysbiosis-Associated Gut Inflammation**				
**Animal Strains**	**GI Phenotype**	**Skeletal Phenotype**	**Systemic and Other Organ Phenotype**	**Reference**
GF mice	serotonin ↑ CD4+ T cells ↓ TNFα ↓	BV/TV, Tb.N. and BFR ↑; Tb.Sp. and OC.N. ↓ BM-CD4+T cells, Th17, TNFα and IL-6 ↓ BM-OC precursor ↓OC fusion ↓	serum OCN, CTX, Ca^2+^, serotonin and TNF ↓ IGF ↑	[88,89]
ConvD-GF	compared with GF mice: colon *Rankl*, *TNFα*, *IL-1β ↑* cecal SCFA ↑	compared with GF mice: femur length↑ Periosteal area ↑ endosteal area ↑ IGF-1 and RANKL in BM ↑	serum CTX-I, PINP: n.s. serum IGF-1 ↑ muscle IGF-1 ↓ fat pad ↑	[90]
**Models Based on Chemical Induction**				
**Animal Strains**	**GI Phenotype**	**Skeletal Phenotype**	**Systemic and Other Organ Phenotype**	**Reference**
TNBS	crypt loss, cellularity, and edema	BV/TV, BFR and osteoid surface ↓ OC.S ↑ % of TNFα+, IL-6+, RANKL+, and OPG+ osteocytes ↓ % of sclerostin+ and IGF+ osteocyte ↑	N/A	[91]
TNBS	Severe colitis; colon TNF, IFNγ, IL-17, IL-1β ↑	cortical bone fraction ↓ BV/TV and Tb.Th ↓	serum OCN and RANKL ↓ serum tDPD, TNFα and IL-6 ↑ renal Ca^2+^ reabsorption ↓ urinal Ca^2+^ excretion ↑ renal TRPV5 and Klotho ↓	[70,92]
DSS	mucosal inflammation and ulceration	BMD, Tb.Th., MAR, and Ct.BMD ↓ OB.Ar ↓ OC.Ar ↑ Growth plate height and cartilage gene ↓	Femoral fat pads ↑ Liver mass ↓ serum TNFα ↑ IGF-1 ↓	[93]
DSS	colonic length ↓	Alveolar bone loss ↑	Liver cystine ↓	[94]
DSS	lymphocyte aggregates ↑; colon TNFα, IFNγ, IL-6 and IL-22 ↑	Trabecular BV/TV, Tb.N., Tb.Th. ↓ Tb.Sp. ↑ growth plate thickness and ColX ↓ BFR and osteoblast surface (OB.S) ↓; TNFα ↑	inguinal fat mass ↓ retroperitoneal fat mass↑ serum OCN ↓	[95]
**Adoptive Transfer of Effector Immune Cells to Immunodeficient Mice**				
**Animal Strains**	**GI Phenotype**	**Skeletal Phenotype**	**Systemic and Other Organ Phenotype**	**Reference**
CD4+IL-10−/−*Rag1−/−* transfer	Severe colitis; epithelial injury; colon TNF, IFNγ, IL-17, IL-1β ↑	Ct.Ar/Tt.Ar ↓ BV/TV, Conn.D, Tb.N, and Tb.Th. ↓	serum tDPD and TNFα ↑ serum RANKL ↓ renal Ca^2+^ reabsorption and TRPV5 ↓ urinal Ca^2+^ excretion ↑	[92]
CD4+CD45RB-*scid/scid-* transfer	Colitis; immune cell infiltration in colon	Osteopenia BMD and osteoblast number (OB.N) ↓ TNFα ↑	BW ↓ SAA and WBC ↑ RBC, haemoglobin, and haematocrit ↓ serum PTH, ALP ↓ TRAP ↑	[96]
CD4+CD45RB-*Rag1−/−* transfer	IBD	%CX_3_CR1+OC ↑ OC from IBD mice induce TNFα-producing CD4+T cells	BW ↓	[97]

Abbreviation: BFR, bone formation rate; BV/TV, Bone volume/Total volume; BMD, bone mineral density; BM, bone marrow; BW, body weight; Conn.D, connective density; Ct.Ar/Tt.Ar, cortical bone fraction; MAR, mineral apposition rate; Oc.S, OC surface; OB, osteoblast; OC, osteoclast; RBC, red blood cell; SAA, serum amyloid; SCFA, short chain fatty acid; TJ protein, Tight junction proteins; Tb. Sp., trabecular spacing; Tb.Th., trabecular thickness; tDPD, deoxypyridinoline; WBC, white blood cell; N/A, not applicable; n.s., no significant difference. ↑ indicates increase, ↓ indicates decrease.

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
