# Peer review of "Mechanisms Underlying Bone Loss Associated with Gut Inflammation"

_ijms, 2019, doi:10.3390/ijms20246323_

Round 1

Reviewer 1 Report

The authors Ke et al. provided a review article on „Mechanisms Underlying Bone Loss Associated with 2 Gut Inflammation”. The review addresses a highly relevant topic, which is frequently underestimated in research.

Major:

Line 63: Sentence referring to the genetic locus is unclear. When referring to SNPs please use a uniform nomenclature. Lines 75/76: are increased OPG levels not inhibiting osteoclastogenesis and bone lass – how should this contribute to bone loss seen in these patients? What about nutrition – patients with IBD often stick to a specific diet – how could this affect the bones? In paragraph 3 the text and table are somehow redundant. Referring to the gut development in early childhood – is there anything known on the influence of breast-feeding? As this is thought to improve gut barrier and muscle function… I sometimes lost track because the different inflammatory gut diseases were mixed and associated with RA and OA phenotypes. To reach a broader audience the text is too long and should be shortened/simplified for understanding. Also the extensive use of abbreviations is sometimes confusing and could reduce the readership. An overview image (images) how the different factors work and affect each other would be helpful for the reader to understand the rather complex content of the manuscript.

Minor:

Sometimes spacing mistakes occurred (especially in front of references). When citing two reference the content between the [] is split. Manuscript should be checked for type-O’s and small grammar mistakes.

Author Response

Response to Reviewers’ Critiques

IJMS- 641366: Mechanisms Underlying Bone Loss Associated with Gut Inflammation

Ke, K. et al.,

Dear Editors:

We thank the reviewers for taking the time to evaluate our manuscript. We have revised the manuscript accordingly and as outlined in the response (in bold) to comments below: (vertical lines on right margins indicate changes in corresponding lines)

Reviewer 1:

Comments and Suggestions for Authors

The authors Ke et al. provided a review article on “Mechanisms Underlying Bone Loss Associated with 2 Gut Inflammation”. The review addresses a highly relevant topic, which is frequently underestimated in research.

Major:

-Line 63: Sentence referring to the genetic locus is unclear. When referring to SNPs please use a uniform nomenclature.

Response: SNPs were given the uniform nomenclature:

Lines 73-74: “Nemetz et al. showed an increased risk of bone loss in IBD patients with IL-1B polymorphism (IL1B-511, rs16944) associated with hyper secretion of IL-1β [18].”

Line 98: “ In UC patients, OPG levels were significantly lower, suggesting that low OPG levels may be associated with bone loss in UC, but not correlated with c.-223C>T (rs2073617) polymorphism in the TNFRS11B gene [23].”

Line 140: “A recent study from Slomski group revealed that, although IBD patients did not display differences in serum 25(OH)D in comparison to control subjects, a protective effect of the VDR gene TaqI (rs731236, c.1057T>C) allele on BMD was seen in IBD patients and controls.”

 -Lines 75/76: are increased OPG levels not inhibiting osteoclastogenesis and bone loss – how should this contribute to bone loss seen in these patients?

Response: Correlation of increased OPG level and bone mass in these IBD patients has been discussed (Lines 84-91): “Moschen et al. paradoxically showed elevated plasma OPG in IBD patients as well as high levels of OPG released from the inflamed colonic mucosa [20]. However, a negative correlation between OPG plasma levels and spine and femoral neck BMD were found in osteoporotic IBD patients. Although plasma levels of soluble RANKL is not changed, an increased numbers of RANKL+ cells were noted in the lamina muscularis of IBD patients [20].”

“It is possible that in these patients, OPG levels are elevated as a compensatory mechanism in order to counteract the effects of pro-inflammatory cytokines. In addition, it is still unclear how local versus systemic effects of RANKL and OPG vary.”

“Therefore, a further understanding of the roles of local and systemic RANKL/OPG on bone loss in the context of IBD is required.”

-What about nutrition – patients with IBD often stick to a specific diet – how could this affect the bones?

Response: Corresponding description has been modified as: (Lines 166-167)

Among IBD patients, reduced BMD or osteopenia is associated with a lack of nutrient intake and defective nutrient utilization, mainly as a result of vitamin deficiency, insufficient calcium uptake and ultimately malabsorption due to destruction of intestinal villi. Surveys on dietary habit and nutrient intake showed lower bone density in malnourished IBD patients [37] especially when undergoing a flare.”

-In paragraph 3 the text and table are somehow redundant.

Response: Contents in Table 1 are simplified.

-Referring to the gut development in early childhood – is there anything known on the influence of breast-feeding? As this is thought to improve gut barrier and muscle function…

Response: Influence of breast-feeding on improvement of gut barrier has been discussed:

“Breastfeeding may allow passage of multiple components, including lactoferrin, whey, TGFβ, and EGF, from mother’s colostrum to infants’ blood. Those components improve infants’ intestinal permeability by promoting TJ protein expression [153], as well as provide easily digestible nutrition for infants. In addition, IgA antibodies found in breast milk have been shown to improve gut microbiota and overall intestinal health, reducing the risk of IBD.”

 -I sometimes lost track because the different inflammatory gut diseases were mixed and associated with RA and OA phenotypes. To reach a broader audience the text is too long and should be shortened/simplified for understanding.

Response: Paragraph on different inflammatory gut diseases-related arthritis phenotypes has been shortened as:

“The microbiome of an individual consists of more than 1,000 microbial species including bacteria and single-celled eukaryote. Homeostatically, the gut microbiota (GM) provides colonization resistance and regulates immune balance bidirectionally within epithelial barrier and tissue environment, to protect the host from invading pathogens [31]. Perturbed balance in microbial composition has been postulated to be associated with compromised immunity within the gut, and an increased susceptibility to the development of enteropathic arthropathies, such as spondyloarthropathy and psoriatic arthritis [32-34]. A recent cross-sectional study displayed an increased abundance in the Clostridiaceae family of bacteria was shared by patients with IBD-associated arthropathy and RA, potentially due to the effects of bowel surgery history [35]. Fecal microbiota study reveals patients with spondyloarthritis (SpA) had a significant abundance of Ruminococcus gnavus, compared with both RA and healthy controls, correlated with a history of IBD in patients [36]. These emerging evidences suggests a potentially common microbial link for inflammatory arthritis and gut inflammation. However, it remains unknown whether enteropathic arthropathies are secondary to inflammation induced by gut dysbiosis. ”

-Also the extensive use of abbreviations is sometimes confusing and could reduce the readership.

Response: “Abbreviations appeared less than three times were removed.”

-An overview image (images) how the different factors work and affect each other would be helpful for the reader to understand the rather complex content of the manuscript.

Response: Fig.1 has been added to illustrate regulation of gut-residing factors in bone loss.

Please see Fig 1 illustration within manuscript.

Fig.1. Networks of gut-immune response in bone loss. Many factors, including microbiota and/or barrier-damaging factors (e.g. DSS), have been validated in triggering gut inflammation, such as IBD. In inflamed gut, epithelial barrier function can be regulated by nuclear receptros (VDR, FXR), inflammatory signals (NF-κB), and various stress stimulators ( ROS, ER stress). Various inflammatory response can be activated in the lamina propria by the infliltraing bacteria and/or local released inflammatory cytokines from epithelium cells. Dysregulation of the balance between mucosal immune cells orchestrate cytokines production and releasing to blood stream. Systemically provided inflammatory cytokines could be transmitted to bone marrow to enable activation of osteoclastogenic immune cells and cytokines, leading to activated bone resportion process. ”       

Minor:

-Sometimes spacing mistakes occurred (especially in front of references).

Response: Spacing mistakes have been corrected.

-When citing two reference the content between the [] is split. Manuscript should be checked for type-O’s and small grammar mistakes.

Response: Split errors on [] have been corrected.

Reviewer 2 Report

The authors provide a comprehensive and well-structured review article summarizing the current knowledge on a clinically relevant topic, i.e gut inflammation-induced bone loss. There are only few issues that may be addressed by the authors in order to further improve their article.

Specific comments:

There is no need to introduce the abbreviation “EIM” for extra-intestinal manifestation, especially since it is only used once. With respect to VDR and microbiota it may be relevant to also refer to a recent GWAS study showing that the VDR locus is associated with microbiome diversity (Wang et al. Nat. Genet., 2016). In section 3.1.11. the reference to “Tobias et al.” is probably wrong and should be “Schmidt et al.”. The subheading for section 4, which is the same as for section 3, is probably wrong, as the section rather refers to potential mechanisms of gut inflammation-induced bone loss. Although this might be quite difficult, it would be advantageous, if the authors could provide a simplified schematic presentation summarizing the main knowledge on gut inflammation-induced bone loss.

Author Response

Response to Reviewers’ Critiques

IJMS- 641366: Mechanisms Underlying Bone Loss Associated with Gut Inflammation

Ke, K. et al.,

Dear Editors:

We thank the reviewers for taking the time to evaluate our manuscript. We have revised the manuscript accordingly and as outlined in the response (in bold) to comments below: (vertical lines on right margins indicate changes in corresponding lines)

Reviewer 2:

Comments and Suggestions for Authors

The authors provide a comprehensive and well-structured review article summarizing the current knowledge on a clinically relevant topic, i.e gut inflammation-induced bone loss. There are only few issues that may be addressed by the authors in order to further improve their article.

Specific comments:

-There is no need to introduce the abbreviation “EIM” for extra-intestinal manifestation, especially since it is only used once.

Response: Abbreviation on “EIM” has been removed.

-With respect to VDR and microbiota it may be relevant to also refer to a recent GWAS study showing that the VDR locus is associated with microbiome diversity (Wang et al. Nat. Genet., 2016).

Response: reference ‘Wang et al. Nat. Genet., 2016’ has been discussed and cited as:

“GWAS analysis of the gut microbiota identified a significant association of bacterial abundance and VDR loci [42].”

-In section 3.1.11. the reference to “Tobias et al.” is probably wrong and should be “Schmidt et al.”.

Response: “Tobias et al.” has been changed to “Schmidt et al.”.

-The subheading for section 4, which is the same as for section 3, is probably wrong, as the section rather refers to potential mechanisms of gut inflammation-induced bone loss.

Response: The subheading for section 4 has been replaced as:

“Networks of gut-residing factors regulating bone marrow microenvironment and bone loss”.

-Although this might be quite difficult, it would be advantageous, if the authors could provide a simplified schematic presentation summarizing the main knowledge on gut inflammation-induced bone loss.

Response: A simplified schematic presentation has been included as Fig.1.
